# Retromode Imaging Modality of Epiretinal Membranes

**DOI:** 10.3390/jcm11143936

**Published:** 2022-07-06

**Authors:** Alfonso Savastano, Matteo Ripa, Maria Cristina Savastano, Tomaso Caporossi, Daniela Bacherini, Raphael Kilian, Clara Rizzo, Stanislao Rizzo

**Affiliations:** 1Ophthalmology Unit, Fondazione Policlinico Universitario Agostino Gemelli IRCCS, 00168 Rome, Italy; asavastano21@gmail.com (A.S.); matteof12@gmail.com (M.R.); tomaso.caporossi@gmail.com (T.C.); stanislao.rizzo@policlinicogemelli.it (S.R.); 2Ophthalmology Unit, Catholic University “Sacro Cuore”, 00168 Rome, Italy; 3Department of Translational Surgery and Medicine, Eye Clinic, University of Florence, Azienda Ospedaliero-Universitaria Careggi, 50134 Florence, Italy; daniela.bacherini@gmail.com; 4Ophthalmology Unit, University of Verona, 37129 Verona, Italy; raphaelkilian8@yahoo.it; 5Ophthalmology Unit, Department of Surgical, Medical, Molecular and Critical Area Pathology, University of Pisa, 56124 Pisa, Italy; clararizzo2@gmail.com; 6Consiglio Nazionale delle Ricerche, Istituto di Neuroscienze, 56127 Pisa, Italy

**Keywords:** epiretinal membranes, retromode retinal imaging, confocal scanning laser ophthalmoscope, fundus autofluorescence, personalized medicine

## Abstract

(1) Purpose: To determine the characteristics of macular epiretinal membranes (ERM) using non-invasive retromode imaging (RMI) and to compare retromode images with those acquired via fundus autofluorescence (FAF) and fundus photography. (2) Methods: Prospective observational case-series study including patients with macular ERM with no other ocular disease affecting their morphology and/or imaging quality. We compared RMI, FAF and fundus photography features by cropping and overlapping images to obtain topographic correspondence. (3) Results: In total, 21 eyes (21 patients) affected by ERM were included in this study. The mean area of retinal folds detected by RMI was significantly higher than that detected by FAF (11.85 ± 3.92 mm^2^ and 5.67 ± 2.15 mm^2^, respectively, *p* < 0.05) and similar to that revealed by fundus photography (11.85 ± 3.92 mm^2^ and 10.58 ± 3.45 mm^2^, respectively, *p* = 0.277). (4) Conclusions: RMI appears to be a useful tool in the evaluation of ERMs. It allows for an accurate visualization of the real extension of the retinal folds and provides a precise structural assessment of the macula before surgery. Clinicians should be aware of RMI’s advantages and should be able to use them to warrant a wide range of information and, thus, a more personalized therapeutic approach.

## 1. Introduction

Epiretinal membranes (ERMs) develop above the internal limiting membrane (ILM) of the retina and represent a relatively common macular finding. Their appearance varies widely from patient to patient, ranging from a translucent wrinkling of the inner retinal surface, all the way to an extensive, thick, epiretinal cellular proliferation [1].

The retinal surgeon benefits greatly from a precise preoperative view of the ERM, which is made possible by continuous advancements in imaging technology.

Thanks to its high-definition cross-sectional scans, optical coherence tomography (OCT) has revolutionized the diagnostic visualization of ERMs [2]. Other common imaging techniques such as fundus autofluorescence (FAF) and enface OCT technology have also been shown to be effective at visualizing ERMs and, in some cases, even at predicting certain post-operative outcomes [3].

The novel retromode imaging (RMI) is a noncontact and a noninvasive imaging method that relies on the newly introduced confocal scanning laser ophthalmoscopic technology. Briefly, there are two kinds of light returning back from the fundus once it is illuminated, a direct reflex and a scattered light. Retromode imaging uses a laterally deviated confocal aperture with a central stop to block the direct light reflex and to collect the backscattered light from one direction [4]. This enables the formation of pseudo-3D images, through the creation of a shadow to one side of the abnormal feature that is being investigated, which eventually enhances the contrast of the lesion. Knowing the ability of infrared lasers to penetrate deeper retinal layers, retromode imaging with infrared laser-technology has been used to evaluate retinal pathological changes in several retinal and choroidal diseases [5].

RMI is performed by a scanning laser ophthalmoscope (SLO) working at an infrared wavelength, which creates a pseudo three-dimensional image with highly contrasting margins of retinal abnormalities (e.g., neurosensory detachments, retinal holes, retinoschisis and intraretinal cystic fluid) [6]. However, to the best of our knowledge, there are no reports describing the retromode imaging of ERMs.

The main outcome of this study was to evaluate morphological features, such as the area of retinal folds and the area of intraretinal cystic spaces, in eyes with ERM. This was achieved by cropping and overlapping images of the ERMs using noninvasive retromode imaging and comparing these to color fundus images. Moreover, we examined the correlation between retromode findings and fundus autofluorescence (FAF).

## 2. Materials and Methods

The study was performed in accordance with the Declaration of Helsinki and was approved by the Ethics Committee (protocol ID number 3680/20). Written informed consent for participation to the research was obtained from the patients after explanation of the purpose and the process of the study.

We performed a monocentric prospective case-series study enrolling patients with an ERM who attended the ophthalmology unit at the Fondazione Policlinico Universitario A. Gemelli IRCCS, Rome, Italy, from 1 January–1 July 2021.

A comprehensive ocular examination, including measurements of the refractive error (spherical equivalent) and the best correct visual acuity, and a dilated macular examination (through indirect ophthalmoscopy) were performed on the day of enrollment. Patients then received a thorough posterior pole investigation through various imaging techniques (i.e., RMI, FAF, color fundus photography and OCT B-scanning). All scans were performed using the Mirante SLO/OCT (Nidek Co, Gamagori, Japan), providing an image field of 40 degrees, an optical resolution of 16 to 20 μm and an image size of up to 1024 × 720 pixels. Inclusion criteria were the presence of an epiretinal membrane confirmed via OCT-B scans with no other concomitant intraocular disease. Eyes with a history of ocular trauma, previous intraocular surgery (with the exception of cataract surgery) or any abnormal intraocular findings (i.e., diabetic retinopathy, neovascular age-related macular degeneration, retinal angiomatous proliferation, angioid streaks, pathological myopia, retinal detachment), were excluded from the study. Images with poor quality due to severe cataracts or unstable fixation were also a reason for exclusion.

After the examinations were performed (1–6 months), all included patients underwent surgery for peeling of the ERM. This was performed by two expert vitreo-retinal surgeons, using 23- or 25-gauge pars plana vitrectomy (PPV) (Constellation Vision System, Alcon, Fort Worth, TX, USA) according to the surgeon’s choice, as already described by Savastano et al. [7].

### 2.1. Images Detection

After completing OCT scans of the entire area within the vascular arcades, the fundus was investigated using fundus photography, FAF and retromode imaging. The OCT acquisition protocol consisted of a 6 × 6 mm (mm) three-dimensional vertical scanning area, centered on the fovea comprising 512 × 128 scans. Color photos from Nidek Mirante had a 45-degree field of view with a resolution of 1024 × 720 pixels. Three separate laser wavelengths (red—670 nm, green—532 nm, blue—488 nm) are combined to create color pictures, which are then connected to a specific sensor for each wavelength. A blue-light excitation wavelength of 488 nm (emission > 500 nm) is used to obtain fundus autofluorescence (FAF), whereas the retromode imaging system uses an infrared laser light (790 nm). A right-deviated aperture and a left-deviated aperture were used to obtain two separate retromode pictures per eye. All the tomographic images were acquired, averaging up to 30 frames per image.

### 2.2. Data Analysis

Image processing was performed using the ImageJ software (National Institutes of Health, MD, USA).

Fundus Autofluorescence, retromode and fundus photography images were cropped and superimposed to have topographic correspondence, and the surface area of each image was measured in square mm^2^.

All images were reviewed separately by two authors to identify, record and interpret the various morphologic alterations associated with ERM. In case of disagreement between the two, a third expert would choose which of the measurements was more accurate.

All images were exported as high-quality tiff files and were collected using the REDCap platform (Vanderbilt University), a secure web application for building and managing online surveys and databases [8].

We matched retromode images with color fundus photographs, FAF images and OCT B-scans using various reference points.

The optic nerve’s superior and inferior margins and retinal blood vessels on color fundus photography (blue lines and orange lines, respectively) were matched with their counterparts identified on retromode images, as shown in Figure 1. On the other hand, while FAF images were matched to retromode images using the retinal blood vessels alone (orange lines Figure 2), for OCT B-scans we used both the neuroretinal rim (grey lines) and the extension of the ERM itself. Particularly, the ERM margins on OCT images were projected onto the retromode image through *orange lines*, whereas black lines parallel to those originating from the neuroretinal rim were used to project the cystic spaces identified on OCT B-scans to their retromode counterparts (Figure 3).

### 2.3. Statistical Analysis

All statistical analyses were performed using SPSS version 27 (IBM-SPSS, Chicago, IL, USA). Quantitative variables were expressed as mean and standard deviation (SD), whereas qualitative variables were displayed as percentages. According to the normality test results, the Student’s *t*-test was used to compare the independent and paired samples. Pearson’s chi-squared test was used to analyze the distribution of the areas of retinal folds among the different multimodal imaging techniques. For each imaging modality, interobserver agreement was calculated using a kappa statistic [9]. The interobserver agreement between the two examiners was not inferior to 0.93 for each comparison. A *p* value of less than 0.5 was considered statistically significant.

## 3. Results

Twenty-one eyes (21 patients) with an epiretinal membrane were included. Twelve patients (57.1%) were male, nine (42.9%) were female and the mean age was 68.38 ± 7.89 years with a range of 51 to 83 years. Snellen best-corrected visual acuity (BCVA) ranged from 20/320 to 20/25.

The mean area of retinal folds in the FAF images was 5.67 ± 2.15 mm^2^ (range 3.12–11.02 mm^2^), whereas those in the fundus photographs and RMI were 10.58 ± 3.45 mm^2^ (range 5.24–15.62 mm^2^) and 11.85 ± 3.91 mm^2^ (range 19.98–11.85 mm^2^), respectively (Table 1).

Retromode images demonstrated areas of increased reflectance with fingerprint patterns containing radiating retinal striae centered on the fovea (blue arrowheads), along with areas of decreased reflectance with more prominent large choroidal vessels in the pericentral area (dark choroid aspect observed in 5 of 21 eyes) (multiple yellow arrowheads) (Figure 4).

The areas of increased reflectance, with fingerprint patterns containing radiating retinal striae centered on the fovea in RMI and the retinal folds in the FAF and fundus photography, are displayed in Figure 5.

FAF images showed an irregular-shaped area of hypo autofluorescence with distorted retinal vessels (green arrowheads). This area corresponded to the extent of the ERM (Figure 4).

Fundus photography, on the other hand, displayed the epiretinal proliferation as a whitish reflex and evident retinal folds in the macular area that were strictly related to distorted retinal vessels in the FAF images and to fingerprint appearance in the retromode images.

Overall, the lesions in the retromode images seemed more extensive than those in the FAF images. However, retromode imaging showed numerous ovoidal or polygonal cystoid spaces located in only two eyes. Both displayed a large cystoid space beneath the fovea and surrounding small cystic spaces for OCT, and the wider the area of exudation was for RMI, the higher the retinal profile was for optical coherence tomography (OCT).

While examining ERM’s pathological findings themselves, in this study we also aimed at investigating the correspondence between retromode images and color fundus photographs. Particularly, we analyzed ERM patterns such as the cystic area (in both retromode and horizontal B-scan images) and the area of retinal folding (on FAF, fundus photograph, retromode, horizontal B-scan). The comparison between the different imaging modalities in the 21 enrolled eyes is shown in Table 2.

There were significant differences between FAF and retro-mode imaging in the detection of retinal folds (i.e., macular traction) (*p* < 0.05).

The mean area of retinal folds detected by RMI was significantly higher than that detected by FAF (11.85 ± 3.92 mm^2^ and 5.67 ± 2.15 mm^2^, respectively, *p* < 0.05), similar to that detected by fundus photography (11.85 ± 3.92 mm^2^ and 10.58 ± 3.45 mm^2^, respectively, *p* = 0.277).

Pearson’s correlation test revealed a positive correlation between the areas of retinal folds detected by RMI and those detected by fundus photographs, FAF and OCT (r = −0.90, *p* < 0.001, r = −0.63, *p* < 0.001, r = −0.35, *p* = 0.003, respectively).

## 4. Discussion

In this study we evaluated the pathological findings related to ERMs using and comparing three imaging techniques: retromode imaging, color fundus photography and fundus autofluorescence.

Thanks to its innovative features, retromode imaging could ease the identification of some ERM features, such as the retinal folds, vessel traction and cystic area, compared to other imaging modalities [4]. The matching between the SD-OCT scans and retromode images of ERMs using pixel-drawing software (ImageJ) has already been described [10]. Indeed, the pseudo-3D effect obtained by filtering the infrared light through a laterally deviated confocal aperture with a central stop creates a directional shadow according to the laterality of the annular aperture and is very effective at detecting retinal folds [4].

Among the classic imaging modalities able to document structural changes caused by ERMs, fundus photography is very useful, as this is a fast, noninvasive and easy test that enables clinicians to gain a direct view of the fundus [11]. Retromode imaging is similar to fundus photography in that it provides a similar field of view and can be performed very quickly in a noninvasive manner. Our comparison between these two techniques indicates a high clinical value of RMI, as it provides a good detection of the involved area. For fundus photography, ERM-related retinal changes are frequently represented by a yellowish/whitish appearance of the membrane associated with vessel tortuosity. retromode imaging can detect the involved macular area in pseudo-3D images with great sensitivity, delivering a more complete view of the anatomical conformation of the retina to the surgeon.

Moreover, monitoring the progression of the hyporeflective areas of the ERM on RMI may be very useful during the follow-up of affected patients, as their enlargement is very easy to detect.

As previously mentioned, retromode images demonstrated areas of increased reflectance with fingerprint patterns, containing radiating retinal striae centered on the fovea. We speculate these may represent areas of splitting of the horizontally oriented internal cone fibers, [12] whereas the areas of decreased reflectance corresponded to the presence of more prominent large choroidal vessels (“dark choroid” aspect). Ahn, S.J. et al. [13] identified a similar finding as well. We observed similar results as in a previous study on myopic patients, but, still, the precise explanation for this remains obscure [12]. One explanation might be that the ERM causes a shadow effect, which obscures the underlying choroid.

RMI enhances the detection quality of ERM’s features compared to other common imaging techniques. In our cohort, this technique allowed to better visualize vessel traction and retinal folds compared to the other imaging modalities (FAF and fundus photography, *p* < 0.05). The ERM surface area detected using retromode was significantly larger than that detected by FAF (*p* < 0.05) and quite similar to that detected by fundus photography (*p* = 0.277). These results suggest that retromode imaging may be useful as a supplementary test for ERM detection.

Even though retromode imaging offers several advantages, it also displays some limitations. Above all, in this study, the detection rate of cystoid spaces on RMI was very low (only 2/21 eyes), whereas horizontal B-scanning revealed these lesions to be present in 100% of patients. In fact, the comparison of lesions in retromode imaging is often difficult, as many fundus details were obscured in these images. Moreover, as with most of the newly introduced imaging techniques, another drawback of RMI is its high cost.

Two major limitations of this study are its limited sample size and the fact that all images were manually cropped and superimposed. By comparing the measurements obtained by two retinal experts however, we feel like the risk of measurement errors was limited to the minimum. In the future, automated systems able to precisely compare different imaging modalities would be of help. Moreover, we did not analyze progression of ERMs’ associated pathological features and whether RMI features could be able to predict surgical results.

## 5. Conclusions

In conclusion, this study showed that retromode imaging is a new effective tool that can be used to gain a more accurate visualization of the real extension of an epiretinal membrane. Retromode analysis in cases of ERM allows for a pseudo-3D visualization of epiretinal changes and a 360-degree visualization around the fovea. There is no other imaging modality that can simultaneously provide all this information. Alongside the current imaging modalities, retromode imaging may serve as a useful supplementary test for the preoperative evaluation of ERMs. Clinicians should be aware of RMI’s advantages and should be able to use them to warrant a wide range of information and, thus, a more personalized therapeutic approach.

## Figures and Tables

**Figure 1 jcm-11-03936-f001:**
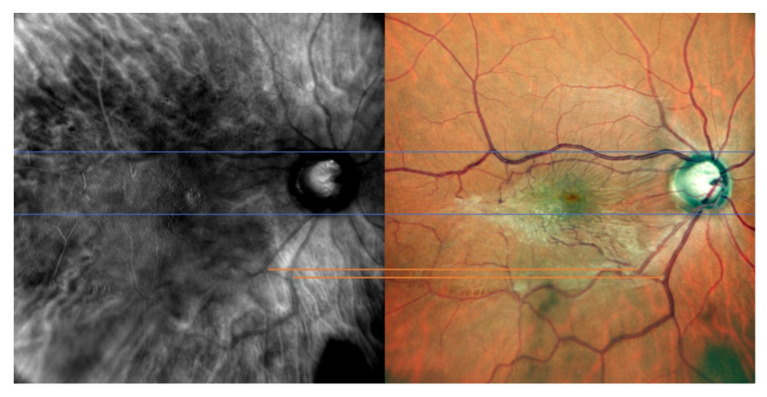
Retromode and fundus photography correspondence: optic nerve vertical outlines’ poles (blue lines) and retinal vessels (orange lines) are used as reference points to check for appropriate image matching.

**Figure 2 jcm-11-03936-f002:**
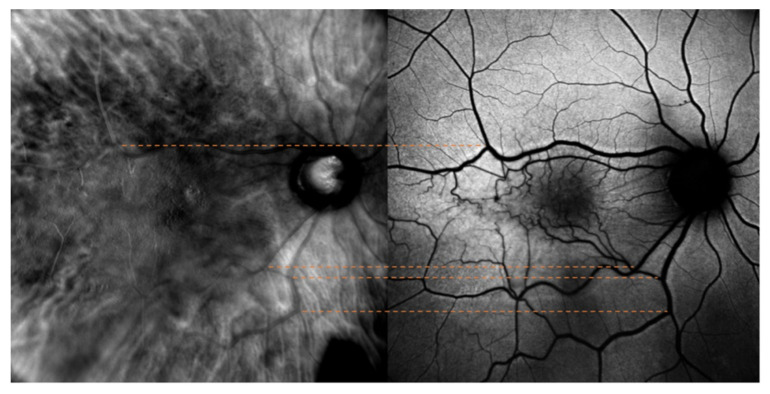
Retromode and fundus autofluorescence (FAF) correspondence: blood vessels on FAF scan (orange lines) are used as reference points to check for appropriate image matching.

**Figure 3 jcm-11-03936-f003:**
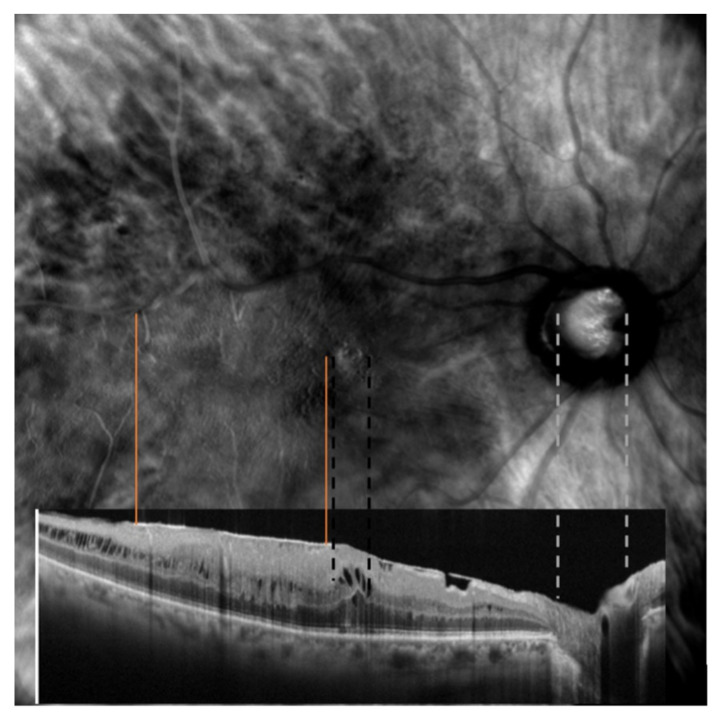
Retromode and SD-OCT correspondence: neuroretinal rim on optical coherence tomography (OCT) scan (grey lines) used as reference points to check for appropriate image matching. Dashed black lines represent the extent of the cystoid spaces.

**Figure 4 jcm-11-03936-f004:**
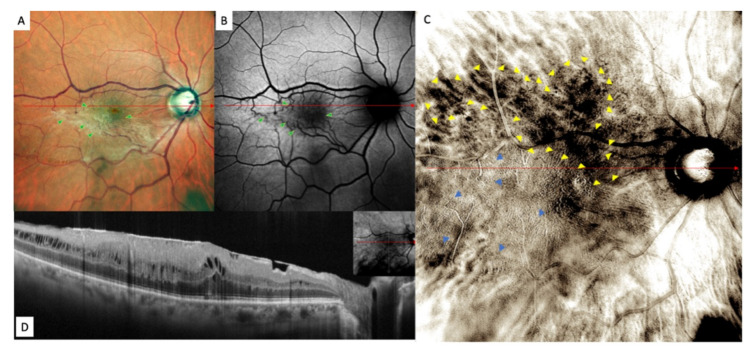
Epiretinal membrane (ERM) in (**A**) confocal scanning laser ophthalmoscopy color fundus photograph (fundus photograph) image, (**B**) fundus autofluorescence (FAF), (**C**) retromode and (**D**) horizontal B-scan. Fundus photography showing the epiretinal proliferation as a whitish light reflex. Retinal folds are well defined in fundus photography and have a fingerprint appearance in retromode imaging analysis. The FAF images showed an irregular-shaped hypoautofluorescence in the macular area.

**Figure 5 jcm-11-03936-f005:**
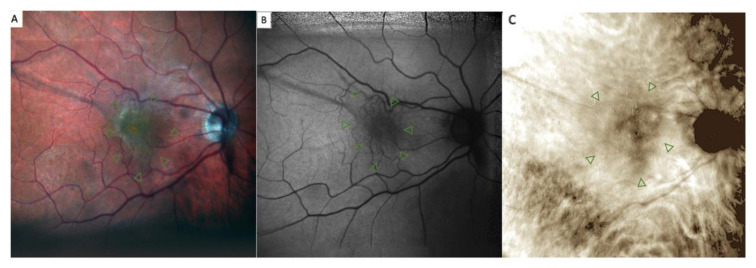
Epiretinal membrane (ERM) in (**A**) confocal scanning laser ophthalmoscopy color fundus photograph, (**B**) fundus autofluorescence (FAF) and (**C**) retromode imaging. The retinal folds are well defined in fundus photography and have a fingerprint appearance in retromode imaging analysis. The FAF image shows an irregular-shaped hypoautofluorescence in the macular area (green arrowheads).

**Table 1 jcm-11-03936-t001:** Demographics and pathological retinal findings of the population of study.

		No. of Patients:	Mean:	SD	Range:
Sex:	Male	12 (57.1%)			
	Female	9 (42.9%)			
Age (years):			68.38	7.89	51–83
FAF folds area (mm^2^)			5.67	2.15	3.12–11.02
Fundus photo folds area (mm^2^)			10.58	3.45	5.24–15.62
RMI folds area (mm^2^)			11.85	3.91	19.98–11.85
B-scan horizontal diameter (µm)			290.52	90.01	100–442
B-scan vertical diameter (µm)			258.57	104.29	100–448
Horizontal B-scan area (µm^2^)			0.79	0.44	0.13–1.98

Abbreviations: RMI: retromode imaging; FAF: fundus autofluorescence; µm: micrometer; SD: standard deviation.

**Table 2 jcm-11-03936-t002:** Summary of data detected by multimodal ERM imaging. FAF: fundus autofluorescence; RMI: retromode imaging.

	Cystic Area RMI	Cystic Area Horizontal B-Scan	Folds Area RMI (mm^2^)	Folds Area FAF (mm2)	Folds Area Fundus Photograph (mm^2^)
#1	no	no	15.61	10.47	14.52
#2	no	no	9.60	4.02	7.30
#3	no	no	14.52	7.01	13.68
#4	no	no	11.01	3.38	10.18
#5	no	no	8.34	4.90	6.22
#6	no	no	13.24	6.22	12.25
#7	2.2	3.7	15.57	7.19	14.84
#8	2.52	4.9	10.06	5.08	9.46
#9	no	no	18.98	7.10	14.66
#10	no	no	8.09	3.98	7.08
#11	no	no	8.22	4.12	7.21
#12	no	no	11.67	5.22	10.97
#13	no	no	15.99	11.02	15.62
#14	no	no	10.36	5.18	10.01
#15	no	no	19.98	7.28	15.61
#16	no	no	8.47	4.23	7.43
#17	no	no	11.24	5.22	11.17
#18	no	no	13.21	6.73	13.74
#19	no	no	9.67	4.38	8.41
#20	no	no	7.09	3.28	6.58
#21	no	no	6.22	3.12	5.24

Abbreviations: RMI: retromode imaging; FAF: fundus autofluorescence.

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
