# Peer review of "Retromode Imaging Modality of Epiretinal Membranes"

_jcm, 2022, doi:10.3390/jcm11143936_

Round 1

Reviewer 1 Report

The methodology of this study is vague and the results are very subjective.

How were the images cropped and superimposed. What was the method used? Manual cropping can be very inaccurate and superimposing manually cropped images is insufficient for a study like this one.

I do not see that the retinal folds are more apparent in retromode images compared to color images. The dark areas in the RMI seem to be cystic spaces which could be better delineated using RMI compared to color but this needs to be better shown using OCT images (preferably using enface OCT images which are probably superior to RMI in detecting cystic spaces too). The authors should also add more images not just one patient.

To sum it up, i doubt any added value using RMI in ERM imaging compared to conventional color photos and en face OCT imaging.

Reviewer 2 Report

Thank you for submitting your paper to this journal. 

RMI is now widely used by clinicians and so there will be a steep learning curve. On some of the images presented, it is difficult to see how an inexperienced clinician in this imaging modality might see more than on the OCT. Would you pls comment on how to train clinicians to enable good quality image analysis for RMI? 

Can you comment on the generalisability of the results to other OCTs? 
